# Microbial Community Structure among Honey Samples of Different Pollen Origin

**DOI:** 10.3390/antibiotics12010101

**Published:** 2023-01-06

**Authors:** Elisavet Stavropoulou, Nikolaos Remmas, Chrysoula (Chrysa) Voidarou, Georgia Vrioni, Theodoros Konstantinidis, Spyridon Ntougias, Athanasios Tsakris

**Affiliations:** 1Department of Microbiology, Medical School, National Kapodistrian University of Athens, 11527 Athens, Greece; 2Department of Environmental Engineering, Democritus University of Thrace, Vas. Sofias 12, 67132 Xanthi, Greece; 3Centre Hospitalier Universitaire Vaudois (CHUV), 1101 Lausanne, Switzerland; 4Department of Agriculture, School of Agriculture, University of Ioannina, 47100 Arta, Greece; 5Laboratory of Hygiene and Environmental Protection, Department of Medicine, Democritus University of Thrace, Dragana, 68100 Alexandroupolis, Greece

**Keywords:** fir and fir–oak honey, *Arbutus unedo* honey, methylotrophs, *Apilactobacillus kun-keei*, *Lonsdalea*, *Zymobacter*

## Abstract

Honey’s antibacterial activity has been recently linked to the inhibitory effects of honey microbiota against a range of foodborne and human pathogens. In the current study, the microbial community structure of honey samples exerting pronounced antimicrobial activity was examined. The honey samples were obtained from different geographical locations in Greece and had diverse pollen origin (fir, cotton, fir–oak, and *Arbutus unedo* honeys). Identification of honey microbiota was performed by high-throughput amplicon sequencing analysis, detecting 335 distinct taxa in the analyzed samples. Regarding ecological indices, the fir and cotton honeys possessed greater diversity than the fir–oak and *Arbutus unedo* ones. *Lactobacillus kunkeei* (basionym of *Apilactobacillus kun-keei*) was the predominant taxon in the fir honey examined. *Lactobacillus* spp. appeared to be favored in honey from fir-originated pollen and nectar since lactobacilli were more pronounced in fir compared to fir–oak honey. *Pseudomonas*, *Streptococcus*, *Lysobacter* and *Meiothermus* were the predominant taxa in cotton honey, whereas *Lonsdalea*, the causing agent of acute oak decline, and *Zymobacter*, an osmotolerant facultative anaerobic fermenter, were the dominant taxa in fir–oak honey. Moreover, methylotrophic bacteria represented 1.3–3% of the total relative abundance, independently of the geographical and pollen origin, indicating that methylotrophy plays an important role in honeybee ecology and functionality. A total of 14 taxa were identified in all examined honey samples, including bacilli/anoxybacilli, paracocci, lysobacters, pseudomonads, and sphingomonads. It is concluded that microbial constituents of the honey samples examined were native gut microbiota of melliferous bees and microbiota of their flowering plants, including both beneficial bacteria, such as potential probiotic strains, and animal and plant pathogens, e.g., *Staphylococcus* spp. and *Lonsdalea* spp. Further experimentation will elucidate aspects of potential application of microbial bioindicators in identifying the authenticity of honey and honeybee-derived products.

## 1. Introduction

The honey market constitutes an important agricultural sector, with the global honey production being estimated at approximately 6.6 billion USD in 2015 [1]. Moreover, honeybees are essential for agriculture, ecology, and the maintenance of life, as the pollination assures the reproduction of plants; thus, beekeeping is actually an issue of concern all around the world [2]. In the European Union (EU), the second largest producer of honey in the globe after China, and EU Member States such as Spain, Romania, Greece, Poland, France, and Italy [3], are the main producers not only of honey, but also of propolis, bee pollen, bee bread, bee wax, and royal jelly (COM(2019) 635 final).

According to latest data from the European Commission, 19 million beehives owned by 615,000 beekeepers produced 275,000 tons of honey in 2020, albeit covering only 60% of the demand in the EU [4], whilst the remaining 40% was covered primarily by Ukraine and secondarily by China [5]. The regulatory framework regarding honey production, properties, and labeling is specified in the Council Directive (EU) 2001/110/EC relating to honey.

In the recent decades, a series of factors, including the extensive application of pesticides, the penetration of invasive species, and deforestation, have added excessive pressure on the global bee population [6]. However, honey is a valuable natural food product, as well as an ingredient used in various personal care products, as well as in traditional medicine. Honey, a hypersaturated sugar solution of high viscosity and osmotic pressure, consists mainly of fructose and glucose, which account for 54.3–87.5% of its content, while minor concentrations of other monosaccharides and oligosaccharides are detected [7]. Moreover, limited amounts of minerals, pollen, amino acids, proteins, organic acids, and their esters are present in honey [7]. Any detected variation can be attributed to the geographical region and flowering, climate conditions, production practices, and storage [7,8]. Moreover, honey has a pH near 4 [9,10] and low water activity, which lies between 0.45 and 0.6 [9,11]. The presence of flavonoids and carotenoids, as well as of minerals, such as Fe, Zn, Cu, and Mn, defines the color of honey [10,12].

In addition to its dietary value, honey is of high medicinal importance since its antimicrobial and wound-healing properties have been noticed since ancient times [13]. Honey’s antibacterial activity, which is linked to enzymatic generation of hydrogen peroxide, wound moisturization ability, and high viscosity, acting as a barrier against infections, are key mode-of-action parameters regarding its medicinal properties [14]. In addition to the generation of hydrogen peroxide via the action of glucose oxidase present in the nectar, the antimicrobial activity and healing properties of honey can be attributed to the antibacterial properties of phenolic acids and flavonoids [14,15]. Moreover, the extremely low water activity and the high osmolarity, as well as the acidic pH, of honey further strengthen the inhibitory effects against bacterial pathogens [16]. On the other hand, spatial and temporal variation in bee pollen and nectar influence the antimicrobial properties of honey [17].

Recently, the antimicrobial activity of honey has also been linked to the inhibitory effects of honey microbiota against foodborne and human pathogens. Pajor et al. [18] investigated the inhibitory effect of bacterial strains isolated from honey against pathogens, reporting that *Bacillus* spp. exhibited antimicrobial activity against *Listeria monocytogenes* ATCC 7644, whereas similar inhibitory activity was also induced by these bacilli against certain *Staphylococcus aureus* and *S. epidermidis* strains. In addition, Voidarou et al. [19] evaluated the therapeutic properties of oregano honey against gastric ulcers and gastritis caused by *Helicobacter pylori*, pointing out that diethyl ether extracts of the honey and the honey itself reduced urease activity exhibited by the specific pathogen. Moreover, Masoura and Gkatzionis [20] examined possible antimicrobial effects of thyme and Manuka honeys against methicillin-resistant *Staphylococcus aureus* (MRSA) strains. The low pH and the high H_2_O_2_ concentration were the key factors influencing the antibacterial activity of monofloral thyme honey. Similarly, Jia et al. [21] also reported the antagonistic action of *Bacillus* sp. A2 from honey against the yeast *Candida albicans* and the bacterial species *Escherichia coli* and *Staphylococcus aureus*. However, employment of high-throughput techniques to uncover microbial diversity in honey is limited. Specifically, Wen et al. [22] performed pyrosequencing to analyze microbial diversity during ripening of vitex honey, reporting the dominance of *Bacillus* spp. and *Lactococcus* spp. and yeasts of the genus *Metschnikowia*.

Thus, the aim of the present study was to identify, for the first time, the predominant microbial communities in *Apis mellifera* honey samples of various pollen origin (fir, cotton, fir–oak, and arbutus) exhibiting high antimicrobial activity against common foodborne and human pathogens, and to comparatively evaluate their microbial community structure and ecological indices, through the application of high-throughput sequencing techniques.

## 2. Results and Discussion

Four honey samples from bees fed with pollen and nectar of various melliferous plant species were examined in terms of their microbial community structure. Regarding diversity indices, the fir and cotton honeys exhibited significantly greater diversity than the fir–oak and *Arbutus unedo* honeys (*a* < *0.05*, in Duncan’s multiple tests for Chao1, Fisher, Shannon, and Simpson diversity indices) (Figure 1). No statistically significant differences were identified between fir and cotton honey, or between fir–oak and *Arbutus unedo* honey regarding all ecological indices estimated, except for the Shannon index, where *Arbutus unedo* honey showed the least score (Figure 1).

The examined fir honey was dominated by lactic acid bacteria of the genus *Lactobacillus* (19.80 ± 0.59% of the total relative abundance), followed by members of the genera *Bradyrhizobium* (11.93 ± 0.20%) and *Pseudomonas* (11.42 ± 1.68%) (Figure 2). Specifically, *L. kunkeei* (current taxonomic name *Apilactobacillus kun-keei*) represented 19.34 ± 0.54% of the total relative abundance at species level in fir honey (Table 1).

Lactobacilli appeared to be favored in honey produced from fir-originated pollen and nectar, with *Lactobacillus* population being more pronounced in fir honey compared to fir–oak honey, indicating a proliferation of this taxon under feeding of honeybees with fir (Figure 3). Interestingly, *Bradyrhizobium* has been recently reported to exert antagonistic activity against various pathogens [23]. Recently, co-existence of lactic acid bacteria and *Bradyrhizobium* resulted in enhanced nodulation with positive impact on plant growth, a fact that may be indicative of co-evolution of these microbiota in certain honeybee hosts [24].

The predominant taxa in cotton honey were *Pseudomonas*, *Streptococcus*, *Lysobacter*, and *Meiothermus*, representing 14.35 ± 2.10%, 7.66 ± 1.69%, 7.07 ± 1.75%, and 6.53 ± 3.93% of the total relative abundance, respectively (Figure 2). *Pseudomonas* is an inhabitant of bee pollen, contributing to the decomposition of pollen walls [25]. As a result, this taxon is part of honeybee microbiome, which is commonly detected in honey [25,26,27]. Notably, streptococci/lactococci, lactobacilli, and enterococci are common microbial constituents of honeybee-collected pollen [28]. Moreover, *Lysobacter* was recently detected as a minor component of intestine honeybee microbiota [29]. Interestingly, *Meiothermus* was reported to be a beneficial bacterium of phytophagous insects, i.e., *Phasmotaenia lanyuhensis* [30].

The major bacterial taxa in fir–oak honey were *Lonsdalea*, causing acute oak decline [31], and *Zymobacter*, a facultative anaerobic fermenter [32], which covered 18.53 ± 4.96% and 17.22 ± 4.48% of the total reads, respectively (Figure 2). Recent findings revealed that “*insects visiting drippy blight diseased red oak trees are contaminated with the pathogenic bacterium Lonsdalea quercina*” [33]; therefore, this bacterium appears to be transmitted from infected oak trees to honeybees and subsequently to fir/oak-originated honey.

The presence of *Zymobacter* may result in a reduced *Lactobacillus* population in these honeybees, since this rarely appearing sugar-tolerant bacterium of *Halomonadaceae* [34] may be favored as a specialized alternative fermenter [35]. Moreover, the acetic acid bacterium *Asaia* is considered as an emerging symbiont of *Apis mellifera* [36]. *Kocuria* spp., which were among the major taxa identified in fir-related honeys (fir and fir–oak honeys; Figure 2), have been identified as the most abundant species in honeybees obtained from beehives in Turkey, which, however, were entomopathogens of honybees [37].

*Lysobacter* (37.31 ± 5.58% of the total relative abundance), *Chryseobacterium* (12.66 ± 4.44%), and *Paenibacillus* (8.81 ± 7.73%) were the predominant microbiota in *Arbutus unedo* honey (Figure 2). Although *Chryseobacterium* is rarely detected in honey [26], this taxon belongs to the native microbiota of *Arbutus unedo* (strawberry tree), since this bacterium was detected in all strawberry tree specimens examined by Martins et al. [38]. Apart from *Lysobacter*, which is a representative of honeybee gut microbiota [29] and a plant-associated microbe with plant-protective properties [39], various *Paenibacillus* spp. have been also detected in the microbiome of honeybee intestine [40]. Although certain members of the genus *Paenibacillus* are entomopathogenic to honeybees [41], various *Bacillus* spp. can be beneficial as biological control agents delivered from honeybees to plants [42].

Among other major taxa (Figure 2), *Sphingomonas* spp., which are also considered as common constituents of honeybee gut microbiota [43,44], were detected in all honey samples examined. Moreover, *Methylobacterium* has been previously detected in *Ceratina* bees [45].

Performance of correlation network analysis in honeys examined showed the strong relationships among certain members of *Actinobacteria* (*Kocuria*, *Nakamurella*, *Tessaracoccus*, *Acidothermus*, *Nocardia*, *Arcanobacterium*, and *Propionicicella*), *Bacteroidota* representatives (*Flavobacterium* and *Ferruginibacter*), the *Rhodobacteraceae* genera *Amaricoccus* and *Gemmobacter*, and the methylotrophic bacterium *Methylotenera*. Such microbiota may be involved in the decomposition of complex compounds present in flowering plants and honey, such as phenolics, flavonoids, lignin, and cellulose [46]. A second distinct cluster was formed solely by the major microbiota of the fir–oak honey (*Lonsdalea*, *Zymobacter*, and lactic acid bacteria such as *Fructobacillus* and *Leuconostoc*), indicating the interaction of *Lonsdalea* with key fermentative taxa in this honey sample.

Lactobacilli and especially *L. kunkeei* (basionym of *Apilactobacillus kun-keei*) strains in the gut of melliferous bees have been reported to induce resistance against deltamethrin, a pesticide that is considered as the major threat for pollinators [47]. *Lactobacillus* spp. are members of the native gut microbiome of honeybees, which have been found to enhance antiviral properties, even during tetracycline treatment that increases sensitivity to viral infections [48]. Moreover, honey with a high *Lactobacillus* population was reported to exhibit the greatest antioxidant activity among other honey samples examined by Wu et al. [49]. *L. kunkeei* (*A. kun-keei*) in the gut of *Apis mellifera* has been reported to exert antimicrobial activity and act as probiotic against the honeybee pathogens *Paenibacillus larvae* and *Melissococccus plutonius* [50,51]. Indeed, *L. kunkeei* is favored by a long evolutionary symbiotic relationship in the gut of honeybees [52]. Moreover, Goh et al. [53] reported that *Lactobacillus* strains, which were isolated from stingless bees and their products, exhibited antimicrobial properties against *Listeria monocytogenes*. Kaškonienė et al. [54] also reported that lactococci and lactobacilli during solid-state lactic acid fermentation of bee pollen exerted antibacterial activity against *Micrococcus luteus*, *Staphylococcus aureus*, and *Escherichia coli*. Similarly, lactic acid bacteria from the intestine tract of honeybees exhibited inhibitory effects against *Bacillus cereus*, *Escherichia coli*, *Pseudomonas aeruginosa*, *Salmonella typhimurium*, and *Staphylococcus aureus* [55].

Comparative analysis of microbial communities in the examined honey samples revealed that their major taxa were either indigenous microbiota of honeybees, mostly beneficial and to a lesser extent potential foodborne and human pathogens, and/or inhabitants of bee pollen and flowering plants, including both plant protective and phytopathogenic microorganisms. In addition, honey originated from cotton (an industrial plant) included a higher proportion of potential foodborne and human pathogens compared to honey originated from bees fed with pollen of wild plants, a fact that may be attributed to the higher abundance of antimicrobial compounds in forest plants [56,57].

A total of 14 taxa were identified in all honey samples (Table 2), independently from their geographic distribution and pollen origin, indicating that these genera may serve as authenticity bioindicators of honey and honeybee-derived products. All of these are either aerobes, capable of growing in low-oxygen conditions (lysobacters and sphingomonads) [58,59] or presenting weak anaerobic growth (*Microbacterium*), as well as being facultative anaerobes (e.g., bacilli/anoxybacilli, paracocci, pseudomonads, staphylococci, and phenylobacteria) or common fermenters of insect gut (*Propionibacterium* spp.). Therefore, restricted oxygen levels in the honeybee gut and/or honey seem to play a role in shaping the microbial community structure in honey.

In all honey samples examined, methylotrophs represented an important fraction of microbial population. In particular, methylotrophs covered 1.3–3.0% of the total relative abundance (Table 3), independently of the geographical and pollen origin. Methanol has been reported to be among the most abundant volatile compounds of honey [60,61]. Interestingly, plant-colonizing methylotrophs of the genus *Methylobacterium* were capable of delivering insecticidal proteins and have been associated with the plant protective properties of the common biocontrol agent *Bacillus thuringiensis* [62]. Moreover, volatiles such as methanol promote the growth of the phyllosphere microbiota [63], serving as an energy source for methylotrophic epiphytes [64]. Thus, both methanol concentration and co-evolution of honeybees with their plant hosts and phyllosphere microbiota appear to shape methylotrophic communities in honey.

In conclusion, native microbiota of the honeybee microbiome, such as fermentative bacteria, either with probiotic properties (e.g., lactobacilli) or potential foodborne and human pathogens (e.g., staphylococci and streptococci), as well as common inhabitants of bee pollen (e.g., pollen wall decomposers, such as *Pseudomonas* spp.) and flowering plants (e.g., beneficial microorganisms of plants and phytophagous insects, such as *Lysobacter* and *Meiothermus* spp., respectively), including plant pathogens of honeybee hosts that are transmitted from flowering plants to honey via honeybees (e.g., *Lonsdalea*, the causing agent of acute oak decline), constitute the microbial community structure in natural honeys.

## 3. Materials and Methods

### 3.1. Collection of Honey Samples

Honey samples from various locations in Greece were obtained aseptically and further examined in terms of their antimicrobial properties. Four honey samples from beehives situated in different geographical areas of Greece, i.e., from the Prefecture of Epirus (the one honey sample was produced from honeybees fed with fir and oak and the other with *Arbutus unedo* pollen and nectar), the Regional unit of Phthiotis, Mendenitsa region (from fir pollen and nectar) and the Regional unit of Karditsa, Palama region (from cotton pollen and nectar), were collected and further studied, due to their high antimicrobial activity (as examined in Stavropoulou et al. [65,66]). These honey samples were subsequently subjected to DNA extraction and phylogenetic analysis of microbial communities through Illumina sequencing.

### 3.2. DNA Extraction from Honey Samples of Different Pollen Origin and Performance of Amplicon Sequencing

The collected honey samples exhibiting high antimicrobial activity against foodborne and human pathogens were subjected to DNA extraction. Genomic DNA was extracted from honey through the use of NucleoSpin Tissue Kit (Macherey-Nagel, Düren, Germany), by following the instructions of the manufacturer. Aliquots of 60 µL of 10 mg/mL lysozyme and 60 µL of 10 mg/mL lysostaphin, as well as 6 µL of 60 U/µL lyticase (enzymes supplied by Sigma-Aldrich, Germany), were added per treated sample to facilitate the lysis of Gram(+) bacteria and yeast strains, respectively. The V4–V5 region of the 16S rRNA gene was amplified using the primers 515F (5′–GTG YCA GCM GCC GCG GTA A–3′) and 909R (5′–CCC CGY CAA TTC MTT TRA GT–3′). No amplification was achieved using primer set ITS1F (5′–CTT GGT CAT TTA GAG GAA GTA A–3′) and ITS4R (5′–TCC TCC GCT TAT TGA TAT GC–3′) for fungi, including yeasts. Amplification of partial 16S rRNA gene was conducted through a thermal scheme comprising of 3 min denaturation at 94 °C, succeeded by 28 cycles of 30 s denaturation at 94 °C, 40 s primer annealing at 53 °C, and 1 min DNA elongation at 72 °C, before a 5 min DNA thermal extension at 72 °C to complete the thermocycling reaction. The amplification reaction for the ITS region was performed by using 2 min denaturation at 94 °C, succeeded by 35 cycles of 30 s denaturation at 95 °C, 30 s primer annealing at 55 °C, and 1 min DNA elongation at 72 °C, before a 10 min DNA thermal extension at 72 °C to complete the thermocycling reaction. Illumina sequencing reactions were carried out at Mr DNA’ (USA) in MiSeq equipment (Illumina, Inc., San Diego, CA, USA) after amplicon cleanup through the use of DNA purification beads.

### 3.3. Bioinformatic Analysis and Analysis of Variance (ANOVA)

Bacterial amplicons were proceeded to demultiplexing and trimming, and partial 16S rRNA gene sequences of abnormal size and low-quality were removed. The assembled reads were further subjected to denoising and chimera discard using USEARCH v.11 [67,68]. Bacterial sequences were clustered and ZOTUs (zero-radius OTU of denoised sequences) were generated, following National Center for Biotechnology Information (NCBI) taxonomy. Ecological indices, i.e., Chao1, Shannon, Simpson, and Fisher diversity indices, were sequentially calculated using the MicrobiomeAnalyst online suite of omics tools [69]. Relationships among bacterial taxa were identified in the MicrobiomeAnalyst platform via multiple correlation network matrix analyses based of the SparCC score at 0.97 correlation coefficient. The sequenced amplicons were deposited in the Sequence Read Archive (SRA) of the NCBI platform under the BioProject accession number PRJNA913297.

Analysis of variance (ANOVA) was conducted using Past v.4.10 [70] to identify statistically significant differences among the relative abundances and diversity indices of the bacterial communities identified in these Greek honey samples exhibiting high antimicrobial activity against foodborne and human pathogens.

## Figures and Tables

**Figure 1 antibiotics-12-00101-f001:**
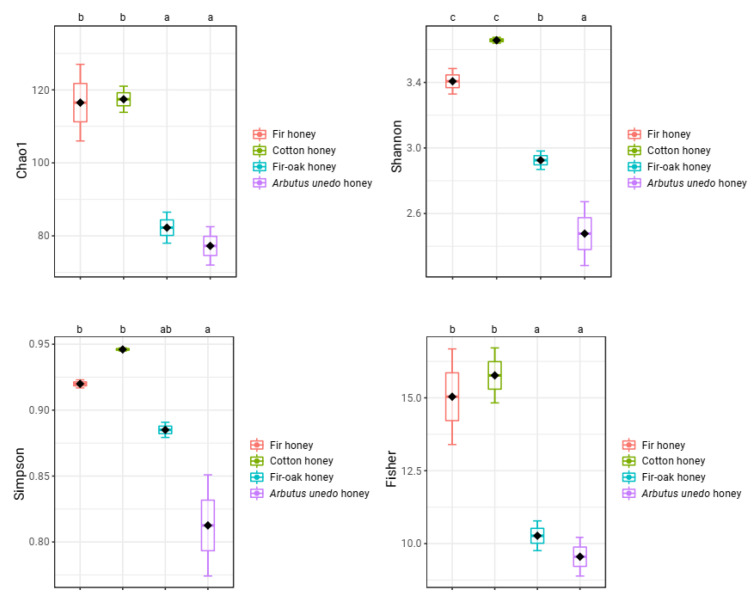
Diversity indices of the examined honey samples. Bars represent ± standard errors of means. Analysis of variance (ANOVA) using Duncan’s multiple post hoc tests at a significance level of 5% (*a* < *0.05*) were carried out to identify statistically significant differences. No letter in common at the top of the subgraphs is indicative of statistically significant differences.

**Figure 2 antibiotics-12-00101-f002:**
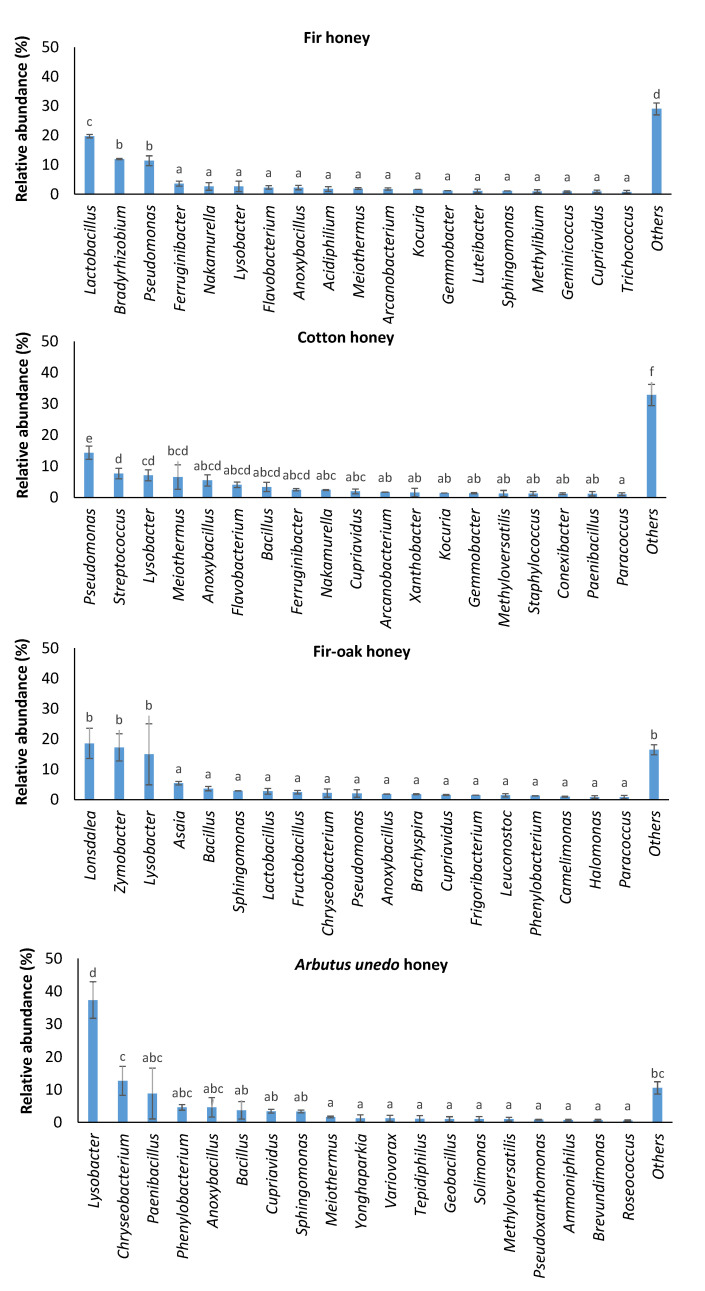
Major bacterial taxa identified in honey samples of different pollen origin. Bars represent ± standard errors of means. Analysis of variance (ANOVA) using Duncan’s multiple post hoc tests at a significance level of 5% (*a* < *0.05*) were carried out to identify statistically significant differences. No letter in common at the top of the bars is indicative of statistically significant differences.

**Figure 3 antibiotics-12-00101-f003:**
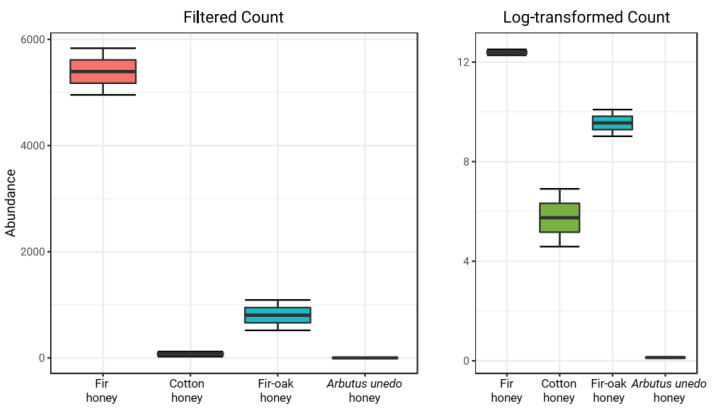
Abundance of *Lactobacillus* genus in the examined honey samples. Bars represent ± standard errors of means.

**Table 1 antibiotics-12-00101-t001:** Relative abundance of *Lactobacillus* species detected in fir honey.

Taxon	Relative Abundance (%)
*Lactobacillus kunkeei*	19.34 ± 0.54
*Lactobacillus johnsonii*	0.31 ± 0.03
*Lactobacillus alvei*	Marginally detected
*Lactobacillus sakei*	Marginally detected
*Lactobacillus mellis*	Marginally detected
*Lactobacillus melliventris*	Marginally detected

**Table 2 antibiotics-12-00101-t002:** List of bacterial taxa identified in all honey samples examined in order to serve as potential authenticity bioindicators of honey.

Taxon ^1^	Fir Honey	Cotton Honey	Fir-Oak Honey	*A. unedo* Honey
*Pseudomonas*	11.42 ± 1.68 (b)	14.35 ± 2.10 (b)	2.02 ± 1.23 (a)	0.32 ± 0.03 (a)
*Lysobacter*	2.67 ± 1.79 (a)	7.07 ± 1.75 (a)	14.93 ± 10.05 (ab)	37.31 ± 5.58 (b)
*Anoxybacillus*	2.24 ± 0.72 (a)	5.47 ± 1.80 (a)	1.78 ± 0.07 (a)	4.60 ± 2.97 (a)
*Bacillus*	0.30 ± 0.30 (a)	3.35 ± 1.47 (a)	3.61 ± 0.76 (a)	3.69 ± 2.66 (a)
*Meiothermus*	1.91 ± 0.29 (a)	6.53 ± 3.93 (a)	0.61 ± 0.07 (a)	1.65 ± 0.26 (a)
*Sphingomonas*	1.14 ± 0.03 (a)	0.63 ± 0.49 (a)	2.86 ± 0.05 (b)	3.33 ± 0.43 (b)
*Cupriavidus*	0.97 ± 0.42 (a)	1.94 ± 0.72 (ab)	1.52 ± 0.18 (ab)	3.41 ± 0.59 (b)
*Phenylobacterium*	0.20 ± 0.06 (a)	0.58 ± 0.23 (a)	1.21 ± 0.03 (a)	4.61 ± 0.87 (b)
*Paracoccus*	0.44 ± 0.05 (a)	1.04 ± 0.43 (a)	0.79 ± 0.54 (a)	0.48 ± 0.23 (a)
*Staphylococcus*	0.35 ± 0.28 (a)	1.23 ± 0.64 (a)	0.23 ± 0.09 (a)	0.57 ± 0.32 (a)
*Methylibium*	1.02 ± 0.48 (a)	0.22 ± 0.06 (a)	0.69 ± 0.12 (a)	0.29 ± 0.03 (a)
*Propionibacterium*	0.65 ± 0.26 (a)	0.61 ± 0.21 (a)	0.27 ± 0.17 (a)	0.17 ± 0.01 (a)
*Methylobacterium*	0.50 ± 0.12 (b)	0.67 ± 0.02 (b)	0.07 ± 0.05 (a)	0.05 ± 0.04 (a)
*Microbacterium*	0.31 ± 0.03 (a)	0.26 ± 0.10 (a)	0.23 ± 0.02 (a)	0.18 ± 0.04 (a)

^1 ^No letter in common within the same row is indicative of statistically significant differences.

**Table 3 antibiotics-12-00101-t003:** Relative abundance of methylotrophic bacteria identified in the examined honey samples.

Taxon	Fir Honey	Cotton Honey	Fir-Oak Honey	*Arbutus unedo* Honey
*Methylibium*	1.02 ± 0.48	0.22 ± 0.06	0.69 ± 0.12	0.29 ± 0.03
*Methylobacterium*	0.50 ± 0.12	0.67 ± 0.02	0.07 ± 0.05	0.05 ± 0.04
*Methylocapsa*	0.18 ± 0.18	n.d.	n.d.	n.d.
*Methylopila*	0.02 ± 0.02	n.d.	0.16 ± 0.16	n.d.
*Methylosinus*	0.25 ± 0.16	n.d.	0.12 ± 0.08	n.d.
*Methylotenera*	0.40 ± 0.01	0.85 ± 0.26	n.d.	n.d.
*Methyloversatilis*	0.60 ± 0.60	1.26 ± 0.99	0.61 ± 0.53	1.00 ± 0.58
Total relative abundance ^1^	2.97 ± 1.20 (a)	3.00 ± 1.18 (a)	1.65 ± 0.44 (a)	1.34 ± 0.59 (a)

^1 ^No letter in common within the same row is indicative of statistically significant differences; n.d., not detected.

## Data Availability

All data of this study are included in this article.

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
