# Peer review of "Microbial Community Structure among Honey Samples of Different Pollen Origin"

_antibiotics, 2023, doi:10.3390/antibiotics12010101_

Round 1

Reviewer 1 Report

The current manuscript describes the prevalence of microbial communities in honey samples in Greece. The paper is well written and the conclusions are supported by the data. Two minor recommendations are below:

1)      In the introduction the authors work to support the importance of honey production and hive numbers by citing articles that are not peer-reviewed. It may be useful to eliminate these citations since they are not included in the references and find other citations to take their place.

2)      In all the Figures (1-3) the legends need to be revised. Specifically there should be a description of the figure in the legend that indicated how significance was identified, number of replicates, etc. Currently the legends are incomplete. In contrast, Table titles should only include a single sentence. Any other important information, such as significant differences, should be identified as a footnote.

Author Response

Dear Reviewer,

Please find below our stepwise response to your comments:

Reviewer#1

Comments and Suggestions for Authors

The current manuscript describes the prevalence of microbial communities in honey samples in Greece. The paper is well written and the conclusions are supported by the data. Two minor recommendations are below:

Response: We would like to thank Reviewer#1.

In the introduction the authors work to support the importance of honey production and hive numbers by citing articles that are not peer-reviewed. It may be useful to eliminate these citations since they are not included in the references and find other citations to take their place.

Response: Following Reviewer’s suggestion, articles that are not peer-reviewed have been omitted from the revised version of the manuscript. Only peer-reviewed articles and official data from European Commission and Eurostat are now included in the revised manuscript.

In all the Figures (1-3) the legends need to be revised. Specifically there should be a description of the figure in the legend that indicated how significance was identified, number of replicates, etc. Currently the legends are incomplete. In contrast, Table titles should only include a single sentence. Any other important information, such as significant differences, should be identified as a footnote.

Response: Following Reviewer’s suggestion, the appropriate additions have been made in Figures 1-3 and in the case of Tables, other important information, such as significant differences, have been moved to footnotes.

Reviewer 2 Report

The present experiment highlighted microbial community structure in honey samples which were previously identified as antimicrobials. The topic is valuable and informative under the current circumstances of increased pesticide burden and the emergence of resistant strains. However, there are some suggestions for further improvement

1.      The title of the manuscript should be changed as it gives the impression that an antimicrobial study is performed in the current experiment.

2.      The abstract requires rephrasing and should emphasize scientific outcomes (Total taxa, sample-specific findings, etc).

3.      In the abstract: Please mention in one sentence the future implication and usefulness of this study.

4.      Line 106-108 avoid writing in points (i, ii).

5.      Please concise the title “3.1. Collection of honey samples exhibiting high antimicrobial activity against foodborne and human pathogens.

6.      Likewise “3.2. DNA extraction from honey samples exhibiting high antimicrobial activity against foodborne and human pathogens and amplicon sequencing”

7.      Modify the caption of figure 1.

8.      Line 134-138: Why the findings of Lactobacillus are correlated with Bradyrhizobium?

9.       Correct the caption of table 3

10.  A concrete discussion is lacking in the manuscript from the perspective of comparative analysis of microbial community observed in four samples. 

Author Response

Dear Reviewer,

Please find below our stepwise response to your comments:

Reviewer#2

The present experiment highlighted microbial community structure in honey samples which were previously identified as antimicrobials. The topic is valuable and informative under the current circumstances of increased pesticide burden and the emergence of resistant strains. However, there are some suggestions for further improvement

Response: We would like to thank Reviewer#2.

  1. The title of the manuscript should be changed as it gives the impression that an antimicrobial study is performed in the current experiment.

Response: The title has been modified to avoid giving the impression of being an antimicrobial study. The modified title is: “Microbial community structure among honey samples of different pollen origin”.

  1. The abstract requires rephrasing and should emphasize scientific outcomes (Total taxa, sample-specific findings, etc).

Response: Abstract has been rephrased and the scientific outcomes have been emphasized in revised version of the manuscript to include total taxa, sample-specific findings, etc. Please see the “Abstract” section of the revised version of the manuscript.

  1. In the abstract: Please mention in one sentence the future implication and usefulness of this study.

Response: Following Reviewer’s suggestion, a sentence for future implication and usefulness of this study has been added in the revised manuscript. Please see the last sentence of the “Abstract” section of the revised version of the manuscript.

  1. Line 106-108 avoid writing in points (i, ii).

Response: Following Reviewer’s suggestion, (i, ii) have been omitted from the revised version of the manuscript.

  1. Please concise the title “3.1. Collection of honey samples exhibiting high antimicrobial activity against foodborne and human pathogens.

Response: “3.1. Collection of honey samples exhibiting high antimicrobial activity against foodborne and human pathogens.” has been changed to “3.1. Collection of honey samples”.

  1. Likewise “3.2. DNA extraction from honey samples exhibiting high antimicrobial activity against foodborne and human pathogens and amplicon sequencing”

Response: “3.2. DNA extraction from honey samples exhibiting high antimicrobial activity against foodborne and human pathogens and amplicon sequencing” has been changed to ““3.2. DNA extraction from honey samples of different pollen origin and performance of amplicon sequencing”

  1. Modify the caption of figure 1.

Response: The caption of Figure 1 has been modified in the revised version of the manuscript.

  1. Line 134-138: Why the findings of Lactobacillusare correlated with Bradyrhizobium?

Response: The following sentence has been added in the revised version of the manuscript:

“Recently, co-existence of lactic acid bacteria and Bradyrhizobium resulted in enhanced nodulation with positive impact on plant growth, a fact that may be indicative of co-evolution of these microbiota in certain honey bee hosts (Kale et al., 2022).”

Kale, N.; Ashwini, M.; Jahagirdar, S.; Shirnalli, G. Potentials of lactic acid bacteria in enhancing nodulation of Bradyrhizobium daqingense and yield in soybean. Legume Res. 2022, 45(4).

  1. Correct the caption of Table 3

Response: The caption of Table 3 has been corrected.

  1. A concrete discussion is lacking in the manuscript from the perspective of comparative analysis of microbial community observed in four samples. 

Response: The following paragraph has been added in the revised version of the manuscript:

“Comparative analysis of microbial communities in the examined honey samples revealed that their major taxa were either indigenous microbiota of honey bees, mostly beneficial and to a lesser extent potential foodborne and human pathogens, and/or inhabitants of bee pollen and flowering plants, including both plant protective and phytopathogenic microorganisms. In addition, honey originated from cotton (an industrial plant) included a higher proportion of potential foodborne and human pathogens compared to honey originated from bees fed with pollen of wild plants, a fact that may be attributed to the higher abundance of antimicrobial compounds in forest plants (BaÄŸci and DiÄŸrak, 1996; Chandra et al., 2017).”

BaÄŸci, E.; DiÄŸrak, M. Antimicrobial activity of essential oils of some Abies (Fir) species from Turkey. Flavour Fragr. J. 1996, 11(4), 251-256.

Chandra, H.; Bishnoi, P.; Yadav, A.; Patni, B.; Mishra, A.P.; Nautiyal, A.R. Antimicrobial resistance and the alternative resources with special emphasis on plant-based antimicrobials - a review. Plants 2017, 6(2), 16.

Reviewer 3 Report

The article describes the microbial community structure of honey samples exerting pronounced antimicrobial activity. Honey microbiota was identified using high-throughput amplicon sequencing analysis and estimation of ecological indices. Microbial constituents of the honey samples were native gut microbiota of melliferous bees and microbiota of their flowering plants, having beneficial and pathogenic strains. The manuscript is well structured.  I think the subject is overall interesting.

Line 148-149: Remove the underline,Moreover, Lysobacter was recently detected as a minor component of intestine honeybee microbiota”

Line 262, 277, 278:  Provide the details of the manufacturer/ supplier of chemicals and equipment e.g., Macherey-Nagel NucleoSpin Tissue Kit, Mr DNA’ (USA), MiSeq 

Author Response

Dear Reviewer,

Please find below our stepwise response to your comments:

Reviewer#3

The article describes the microbial community structure of honey samples exerting pronounced antimicrobial activity. Honey microbiota was identified using high-throughput amplicon sequencing analysis and estimation of ecological indices. Microbial constituents of the honey samples were native gut microbiota of melliferous bees and microbiota of their flowering plants, having beneficial and pathogenic strains. The manuscript is well structured.  I think the subject is overall interesting.

Response: We would like to thank Reviewer#3.

Line 148-149: Remove the underline, “Moreover, Lysobacter was recently detected as a minor component of intestine honeybee microbiota”

Response: The “underline” in Lysobacter has been removed in the revised version of the manuscript.

Line 262, 277, 278:  Provide the details of the manufacturer/ supplier of chemicals and equipment e.g., Macherey-Nagel NucleoSpin Tissue Kit, Mr DNA’ (USA), MiSeq 

Response: Following Reviewer’s suggestion, details on the manufacturer/ supplier of chemicals and equipment used in this study are now provided in the revised version of the manuscript.

Round 2

Reviewer 2 Report

The manuscript is significantly improved based on suggestions.